# Protective Effects of Glucose-Related Protein 78 and 94 on Cisplatin-Mediated Ototoxicity

**DOI:** 10.3390/antiox9080686

**Published:** 2020-08-02

**Authors:** Junyeong Yi, Tae Su Kim, Jhang Ho Pak, Jong Woo Chung

**Affiliations:** 1Department of Otorhinolaryngology-Head and Neck Surgery, University of Ulsan College of Medicine, Asan Medical Center, 88 Olympic-ro 43-gil, Songpa-Gu, Seoul 05505, Korea; junyi0403@gmail.com; 2Department of Otorhinolaryngology, School of Medicine, Kangwon National University, Gangwondaehakgil, Chuncheon, Gangwon-Do 24341, Korea; kimtaesu77@gmail.com; 3Department of Convergence Medicine, University of Ulsan College of Medicine and Asan Institute for Life Sciences, Asan Medical Center, 88 Olympic-ro 43-gil, Songpa-Gu, Seoul 05505, Korea

**Keywords:** cisplatin, ototoxicity, mild ER stress condition, GRP78, GRP94, auditory cells

## Abstract

Cisplatin is a widely used chemotherapeutic drug for treating various solid tumors. Ototoxicity is a major dose-limiting side effect of cisplatin, which causes progressive and irreversible sensorineural hearing loss. Here, we examined the protective effects of glucose-related protein (GRP) 78 and 94, also identified as endoplasmic reticulum (ER) chaperone proteins, on cisplatin-induced ototoxicity. Treating murine auditory cells (HEI-OC1) with 25 μM cisplatin for 24 h increased cell death resulting from excessive intracellular reactive oxygen species (ROS) accumulation and caspase-involved apoptotic signaling pathway activation with subsequent DNA fragmentation. GRP78 and GRP94 expression was increased in cells treated with 3 nM thapsigargin or 0.1 μg/mL tunicamycin for 24 h, referred to as mild ER stress condition. This condition, prior to cisplatin exposure, attenuated cisplatin-induced ototoxicity. The involvement of GRP78 and GRP94 induction was demonstrated by the knockdown of GRP78 or GRP94 expression using small interfering RNAs, which abolished the protective effect of mild ER stress condition on cisplatin-induced cytotoxicity. These results indicated that GRP78 and GRP94 induction plays a protective role in remediating cisplatin-ototoxicity.

## 1. Introduction

Hearing loss, also known as hearing impairment, is generally classified into conductive and sensorineural hearing loss. The latter is caused by several risk factors, including acoustic trauma, aging, ototoxic drug use, autoimmune disease, infection, and genetic disorders. Hearing loss is commonly associated with the loss of auditory hair cells in the cochlea, which is irreversible due to its inability to regenerate irreparable hair cell damage [1]. In particular, various commonly used drugs have ototoxic properties that damage the cochlea or auditory nerve and vestibular system and are referred to as drug-induced hearing loss (DIHL). The ototoxic side effects of drugs, such as salicylates, aminoglycosides, and cisplatin, are bilaterally symmetric or asymmetric, with one ear being affected later. DIHL may arise during or after the end of therapy and may be occasionally recoverable if the drug is immediately discontinued or if the initial damage is allowed to repair. However, further accumulation of ototoxic medication may lead to permanent destruction of the sensory hair cells and, concomitantly, permanent hearing loss [2].

Cisplatin is a chemotherapeutic drug that is extensively used to treat several human solid cancers. Its dose-limited use contributes to numerous adverse effects, and one of them is ototoxicity, which currently cannot be cured or prevented. Cisplatin-induced hearing loss is bilateral, progressive, and irreversible, which is often accompanied by tinnitus and vertigo. This ototoxicity occurs in at least major tissues in the cochlea, including the organ of Corti, striavascularis, and spiral ganglia, mainly damaging the outer hair cells (OHCs) of the organ of Corti [3,4]. The primary target of cisplatin is DNA whose lesion results from the formation of interstrand and intrastrand crosslinks. These cisplatin–DNA adducts consequently lead to cell cycle arrest by blocking DNA synthesis, inhibiting RNA transcription, and ultimately activating pro-apoptotic genes. In addition, cisplatin promotes intracellular reactive oxygen species (ROS) accumulation, which is caused by the depletion of reduced glutathione (GSH), inactivation of antioxidant enzymes, and disturbance of the mitochondrial respiratory chain complex [5,6]. Cisplatin enhanced ROS generation in the cochlear tissue explants of guinea pigs, as directly detected through electron paramagnetic resonance spectrometry [7]. Cisplatin-mediated ROS generation is dependent on the induction and activation of the transient receptor potential vanilloid 1 and an NADPH oxidase isoform, NOX3, in rats and organ of Corti hair cell cultures [8]. It has been further reported that excessive ROS generation through NOX1 and NOX4 activation contributes to cisplatin-induced cochlear injury [9]. Whether protective agents interfere with the anti-tumor effect of cisplatin is a major concern, although antioxidant compounds are used to scavenge cisplatin-triggered ROS accumulation, thus blocking downstream cell death cascade. Therefore, there is an imperative need for therapeutic regimes that ameliorate cisplatin-induced ototoxicity.

Glucose-regulated proteins (GRPs) are chaperone proteins in the endoplasmic reticulum (ER), which play critical roles in maintaining ER homeostasis and assisting nascent polypeptide folding and degradation. Upon ER stress caused by accumulated misfolded proteins and/or imbalanced Ca^2+^, the unfolded protein response (UPR) pathway induces the expression of ER chaperons, thereby restoring the ER to its normal physiologic state. The most abundant glycoproteins in ER chaperons are GRP78 (BiP) and GRP94 (GP96) that enhance proper protein-folding capacity and target misfolded proteins for degradation [10]. Pretreatment with known ER stress-inducing reagents, such as thapsigargin (TG) and tunicamycin (TM), increased GRP78 and GRP94 expression, which rendered LLC-PK1 renal epithelial cells resistant to cell injury and death caused by subsequent exposure to a toxicant, whereas decreased GRP78 expression with antisense oligonucleotides resulted in augmented cytotoxicity [11]. Reduced expression of GRP78 and GRP94 in hepatocarcinoma cells (HepG2) overexpressing cytochrome P450 2E1 was shown to increase susceptibility to oxidative and toxic ER stress [12]. Moreover, GRP78 and GRP94 are involved in the preconditioning effect of ER stress on nephrotoxin- or methylmercury-induced cytotoxicity and ischemia/reperfusion heart injury [13,14,15]. Regarding the physiologic functions of GRP78 and GRP94 in the auditory system, the protective properties of GRP78 against hearing loss have been reported in age-related and noise-induced hearing-impaired animal models [16,17]. However, the protective mechanism of these proteins in acoustic trauma remains unclear.

In this study, we examined the protective effects of GRP78 and GRP94 on cisplatin-induced ototoxicity. Mild ER stress condition in inner ear sensory hair cells (HEI-OC1) resulted in the induction of GRP78 and GRP94 expression, which subsequently attenuated cisplatin-mediated ototoxic side effects, including cytotoxicity, excessive ROS generation, upregulation of apoptosis-related protein expression, and accumulation of DNA fragmentation. Using the small interfering (si) RNA to GRP78 and GRP94, we confirmed that they played crucial roles in protecting hair cells from cisplatin-induced ototoxicity.

## 2. Materials and Methods

### 2.1. Materials

Cell culture medium components were purchased from Life Technologies (Grand Island, NY, USA) unless otherwise indicated. Cisplatin, thapsigargin (TG), and tunicamycin (TM) were purchased from Sigma–Aldrich (St. Louis, MO, USA). Polyclonal antibodies against Bax (556467)and GRP78 (610978) were obtained from BD Biosciences (San Jose, CA, USA); polyclonal antibodies against β-actin (A5441) were obtained from Sigma-Aldrich; polyclonal antibodies against Bcl-2 (3498), caspase-3 (9662), cleaved caspase-3 (9661), and poly (ADP-ribose) polymerase (PARP; 9542) were obtained from Cell Signaling Technology (Danvers, MA, USA); polyclonal antibodies against caspase-7 (sc-81654) were purchased from Santa Cruz Biotechnology (Santa Cruz, CA, USA); polyclonal antibodies for GRP94 (ADI-SPA-850) were purchased from Enzo Life Sciences (Farmingdale, NY, USA). Horseradish peroxidase (HRP)-conjugated secondary antibodies were purchased from Bethyl Laboratories (Montgomery, TX, USA).

### 2.2. Cell Culture

The HEI-OC1 cell line was kindly provided by Dr. Federico Kalinec (Dept. of Cell and Molecular Biology, House Ear Institute, Los Angeles, CA, USA). The HEI-OC1 cells were cultured in high-glucose Dulbecco’s modified Eagle’s medium (DMEM) with 10% fetal bovine serum (FBS) at 33 °C in 10% humidified CO_2_ atmosphere.

### 2.3. Cytotoxicity Assay

Cell viability was evaluated using a colorimetric D-Plus™ CCK cell viability assay kit (Dongin LS, Seoul, Korea), according to the manufacturer’s instructions. The cells were seeded on 96-well plates at a density of 4 × 10^3^ cells/well and grown for 24 h under standard conditions. These cells were exposed to different concentrations of cisplatin, TG, and TM for 24 h. For inducing GRP expression, the cells were pretreated with 3 nM of TG or 0.1 μg/mL of TM for 24 h, followed by the treatment of 25 μM of cisplatin for 24 h. The amount of formazan dye generated was determined by measuring the absorbance at 450 nm using a microplate spectrophotometer (Molecular Devices Corp., Sunnyvale, CA, USA). The absorbance values were converted to percentages for comparison with untreated controls.

### 2.4. Immunoblotting

The cells were washed with ice-cold PBS and lysed with RIPA buffer (Sigma–Aldrich) supplemented with complete protease inhibitor cocktail on ice for 30 min. The supernatants were collected by centrifugation at 13,000× *g* for 20 min, and protein concentrations were determined using a BCA Protein Assay kit (Thermo Fisher Scientific, Waltham, MA, USA). Total soluble proteins (10–30 μg) were separated on 12% sodium dodecyl sulfate polyacrylamide gel and transferred to nitrocellulose membranes (GE Healthcare Biosciences, Uppsala, Sweden). The membranes were blocked using 5% skim milk in TBS-T (10 mM Tris-HCl, pH 7.4, 100 mM NaCl, and 0.1% Tween 20) for 1 h at room temperature. The membranes were probed with their corresponding primary antibodies, followed by the appropriate HRP-conjugated secondary antibodies. Then, immunoreactive bands were detected using enhanced chemiluminescence assay technique (ECL; Dongin LS) and quantified using the ImageQuant LAS 500 biomolecular imager (GE Healthcare Biosciences).

### 2.5. Measurement of Intracellular ROS Production

Intracellular ROS level was measured using a fluorescent dye, 5-(and-6)-chloromethyl-2′,7′-dichlorodihydrofluorescein diacetate acetyl ester (CM-H_2_DCFDA; Molecular Probes, Inc., Eugene, OR, USA). The cells grown on 96-well plates were pre-incubated with 3 nM of TG or 0.1 μg/mL of TM for 24 h and then treated with 25 μM of cisplatin for another 24 h. The cells were then washed twice with Hank’s balanced salt solution (HBSS) and incubated with 5 μM CM-H_2_DCFDA for 20 min at 33 °C in the dark. After washing twice with HBSS, the samples were immediately observed at 485 nm excitation and 535 nm emission using a PerkinElmer VICTOR 3 luminescence spectrometer (Perkin-Elmer, Waltham, MA, USA).

### 2.6. Detection of Apoptosis Using TUNEL Assay

Apoptosis was detected using both LIVE/DEAD Viability/Cytotoxicity Kit (Molecular Probes, Inc., Eugene, OR, USA) and In Situ Cell Death Detection Kit, TMR red (Roche Diagnostics, Indianapolis, IN, USA), according to the manufacturer’s instructions, with slight modifications. The cells grown on the glass coverslip in 6-well culture dishes were pretreated with 3 nM TG or 0.1 μg/mL TM for 24 h and further incubated with 25 μM of cisplatin for 24 h. Live cells were labeled with calcein AM, briefly washed with PBS, and fixed with 4% paraformaldehyde. Then, the cells were permeabilized with 0.1% Triton X-100 in 0.1% sodium citrate for 5 min and incubated with the TUNEL reaction mixture containing terminal deoxynucleotidyl transferase and tetramethyl-rhodamine-dUTP. The cells were examined using the appropriate filter of an Olympus IX71 fluorescence microscope, green fluorescence (ex/em ≈495/≈515 nm) for live cells, and red fluorescence (ex/em ≈495/≈635 nm) for apoptotic cells. The percentage of TUNEL-positive cells was determined by counting ≈1000 cells selected from 3–4 randomly chosen fields of the cover slip.

### 2.7. Transfection with siRNA

The siRNAs of GRP78, GRP94, and scrambled oligonucleotide, as a negative control, were obtained from Genolution Pharmaceuticals, Inc. (Seoul, Korea). The cDNA sequences of GRP78 (GenBank accession number; NM_001163434.1) and GRP94 (GenBank accession number; NM_011631.1) to design the respective siRNAs were as follows: 5′-GAAU GAAUUGGAAAGCUAUUU-3′ for GRP78 and 5′-CUGGAAAUGAGGAGUUAACUU-3′ for GRP94. For the scrambled siRNA, it was 5′-CCUCGUGCCGUUCCAUCAGGUAGUU-3′. The cells were seeded on 24-well culture plate and transiently transfected with each siRNAs (60 nM) using G-fectin (Genolution Pharmaceuticals, Inc., Seoul, Korea), according to the manufacturer’s instructions. Each transfection procedure was performed in quadruplicate. After 24 h, the transfection mixture was replaced with fresh culture medium and further incubated for 2 d. Each transfectant was treated with TG (3 nM) or TM (0.1 μg/mL) for 24 h and incubation with 25 μM of cisplatin for another 24 h. Cell viability and ROS accumulation level were evaluated as described in the previous text.

### 2.8. Statistical Analysis

Data were expressed as means ± standard error (SE) of three independent experiments. Differences between groups were evaluated using Student′s *t*-test or one-way analysis of variance (ANOVA), as appropriate. A *p* value of <0.05 was considered statistically significant.

## 3. Results

### 3.1. Cisplatin-Induced Apoptosis in HEI-OC1 Cells

The HEI-OC1 cells were exposed to different concentrations of cisplatin (5–100 μM) for 24 h to determine the adequate cytotoxic cisplatin concentration, and cell viability was monitored using CCK assay. The cisplatin treatment decreased cell viability in a dose-dependent manner, with a lagging dose between 10 and 15 μM. At 25 μM cisplatin concentration, cell viability was 44.8%, compared with that of the untreated control (Figure 1A). As a result, 25 μM cisplatin concentration was used in our subsequent studies, since this concentration and timepoint were within the range of an estimated half-maximal cytotoxic dose (IC_50_).

It is well established that cisplatin-induced cytotoxicity is closely associated with excessive generation of ROS and activation of apoptosis-related proteins [18,19]. Therefore, we initially measured the levels of cisplatin-induced intracellular ROS using a peroxide-sensitive fluorescent probe, CMH_2_DCFDA. As shown in Figure 1B, DCF fluorescence intensity from the cisplatin-treated cells was five-fold higher than that of the untreated control. Next, we evaluated the changes in expression levels of proteins involved in apoptotic pathways to investigate whether cisplatin-induced cytotoxicity was associated with apoptosis. Immunoblot analyses showed that the expression levels of two mitochondrial proteins, namely Bcl-2 (anti-apoptotic protein) and Bax (pro-apoptotic protein), were contrasting in cisplatin-treated cells, that is, the ratio of Bcl-2/Bax was 1.0 in the untreated control versus 0.24 in cisplatin-treated cells. Moreover, catalytically activated forms (cleaved) of caspase-3 and caspase-7 had a six-fold increase in cisplatin-treated cells, with concomitant cleaved (inactivated) PARP fragment accumulation (Figure 1C). Taken together, these results indicated that excessive ROS accumulation and apoptosis contributed to cisplatin-mediated ototoxicity in HEI-OC1 cells.

### 3.2. Effects of ER Stress Inducers on GRP78 and GRP94 Expressions in HEI-OC1 Cells

The induction of GRP78 and GRP94 expression during ER stress are reported to function in maintaining ER homeostasis, assisting in proper protein folding, and degrading misfolded proteins through chaperone formation [20,21]. The involvement of GRPs in cell survival prompted us to examine the protective roles of GRP78 and GRP94 in cisplatin-mediated ototoxicity. Cell viability was evaluated in TG- or TM-treated cells at various concentrations. Moreover, the 24-h exposure revealed that both inducers’ cytotoxicity was increased dose-dependently. At 3 or 5 nM TG concentration, cell viability was decreased by 92.6% or 90.1%, whereas at 0.05 or 0.1 μg/mL TM concentration, cell viability was 91.2% or 89.2% (Figure 2A,B), indicating that the cytotoxic effects of TG or TM is relatively mild at these concentrations. Treatment with 3 or 5 nM of TG induced significant increases in GRP78 and GRP94 expressions; that is, three-fold and six-fold for GRP78 and GRP94 at both concentrations, respectively. Treatment with 0.05 μg/mL TM resulted in two-and-a-half-fold increase in GRP78 expression, but not in that of GRP94. The expressions of both proteins were significantly increased at 0.1 μg/mL (Figure 2C). Therefore, the aforementioned concentration of TG or TM (3 nM or 0.1 μg/mL, respectively) was used to examine the protective effects of GRP78 and GRP94 on cisplatin-induced ototoxicity.

### 3.3. Protection of GRP78 and GRP94 Induction from Cisplatin-Mediated Ototoxicity

To further examine whether the upregulation of GRP78 and GRP94 attenuated cisplatin-induced cytotoxicity, these proteins were induced by pre-incubating HEI-OC1 cells with 3 nM of TG or 0.1 μg/mL of TM for 24 h, and then exposed to 25 μM of cisplatin for another 24 h. As shown Figure 3A, the CCK assay showed that pretreatment with TM or TG increased cell viability by 29.4% or 27.8% more than that of cisplatin alone. We evaluated the level changes of cisplatin-induced intracellular ROS generation in cells pretreated TM or TG. Cisplatin-triggered ROS accumulation was decreased by 2.9 or 2.2 times, respectively, in cells pretreated with TM or TG, as determined by DCF fluorescence intensity analysis (Figure 3B). When the cells were treated with TM or TG for 24 h, the ROS levels were slightly increased, but the values were significantly lower than that of cisplatin treatment alone (data not shown). This result indicated that GRP induction attenuated cisplatin-triggered intracellular ROS accumulation. Next, we investigated changes in the levels of protein expressions involved in apoptotic pathways, using immunoblot analysis (Figure 3C). Cisplatin treatment did not cause any changes in GRP78 and GRP94 expression by themselves. The ratio of Bcl-2/Bax was 0.13 in cisplatin-treated cells, whereas this ratio was elevated in cells pretreated TM or TG (0.6 or 0.67). In addition, the augmented activation of caspase-3 and caspase-7, as well as cisplatin-induced PARP inactivation dramatically declined in cells pretreated with TM or TG.

Finally, the protective effects of GRP overexpression on cisplatin-induced apoptosis was confirmed through calcein AM staining (green) of viable cells, following the detection of DNA fragmentation using the TUNEL assay method (red). This allowed the researchers to calculate the percentage of apoptotic cells over viable cells. As shown in Figure 4, the percentage of TUNEL-positive cells was 35% in cisplatin-treated cells, whereas pretreatment of TM or TG resulted in a dramatic reduction of this percentage by 8% or 13%, respectively. Taken together, these results indicate that GRP pre-induction inhibits cisplatin-mediated apoptotic events in HEI-OC1 cells, such as oxidative stress, caspase-dependent pathway activation, and dysregulation of apoptosis-regulating mitochondrial proteins.

### 3.4. Effect of GRP78 or GRP94 Knockdown (KD) on Cisplatin-Mediated Ototoxicity

To further validate the protective roles of GRP78 and GRP94 against cisplatin-induced ototoxicity, the pre-induction of GRP78 and GRP94 in TM- or TG-treated cells was inhibited through small interfering (si) RNA transfection, and then the changes in cisplatin-mediated cytotoxicity and ROS accumulation were evaluated. After 72 h of transfection, GRP78 and GRP94 expression levels in both KD transfectants were markedly decreased 0.4 times, compared with that in the scrambled siRNA transfectant, as per the results of the immunoblot analysis. The slight reduction of GRP94 or GRP78 expression observed in GRP78 or GRP94 KD cells was not statistically significant (Figure 5A). Cisplatin-induced cytotoxicity was not changed in either GRP78 or GRP94 KD transfectant, whereas the rescue effect of the TG or TM pretreatment on cell viability was markedly decreased by 30%, compared with that of the scrambled siRNA transfectant (Figure 5B). Concomitantly, each pretreatment further increased cisplatin-triggered ROS accumulation in KD cells (Figure 5C). These results demonstrated that GRP overexpression plays a crucial role in attenuating cisplatin-mediated ototoxicity.

## 4. Discussion

Excessive free radical formation in the cochlea caused by aging, noise exposure, and ototoxic compounds results in sensory hair cell injury, which subsequently leads to hearing loss. Potential free radical generators in the ear include mitochondria, enzymatic reactions, NOX3, and increased intracellular calcium concentration that leads to overproduction of neurotransmitters, such as nitric oxide (NO) and glutamate [2,22,23]. In this respect, maintaining redox homeostasis is crucial in protecting the cochlea and central auditory system against oxidative stress-mediated acoustic trauma. In the present study, we found that pre-induction of GRP78 and GRP94 attenuated the cisplatin-induced ROS accumulation, which protected the HEI-OC1 cells from oxidative injury.

Cisplatin is an effective, widely used anticancer drug; however, its major side effect is ototoxicity with subsequent sensorineuronal hearing loss after high-dose treatment. Cisplatin ototoxicity is known to be associated with at least two mechanisms; DNA adduct formation and ROS accumulation in both the cochlea and the vestibular system, leading to the death of sensory cells through apoptosis or necrosis [24]. For example, cisplatin was found to induce apoptosis HEI-OC1 cells and the neonatal rat organ of Corti explants, which was mediated by ROS generation and lipid peroxidation [25]. Intraperitoneal cisplatin evoked a hearing threshold shift and an intrinsic apoptotic pathway within rat cochleae, which involved the activation of caspase-3 and caspase-7, and modulation of two mitochondrial protein expressions (increased Bax and decreased Bcl-2 levels) [26]. It has been also reported that cisplatin ototoxicity in HEI-OC1 cells is mainly associated with the mitochondrial apoptotic pathway through the activation of ROS/JNK signaling cascade [27]. Consistent with these findings, the present study showed that intracellular ROS accumulation and intrinsic apoptotic pathways mainly contributed to cisplatin-induced cell death in HEI-OC1 cells (Figure 1). Activation of caspase-3 and caspase-7, inactivation of PARP, and altered expression of Bcl-2/Bax were found to be associated with increased levels of DNA fragmentation (Figure 4).

GRP78 and GRP94 induction ensures proper protein folding in the ER, thereby protecting cells from ER dysfunction caused by nutrient deprivation, chemical toxicity, changes in calcium mobilization, oxidative stress, or glycosylation disturbances [28,29]. These were induced by treating the cells with a specific inhibitor of ER Ca^2+^-ATPase (TG) or an N-linked glycosylation inhibitor (TM), which disrupts ER calcium homeostasis or prevents post-translational protein maturation, respectively. The protective mechanisms of GRPs are involved in suppressing intracellular ROS accumulation and stabilizing mitochondrial function [11,30]. In the present study, a dose-dependent cytotoxicity was observed in HEI-OC1 cells exposed to different concentrations of TG or TM for 24 h. At 3 nM of TG or 0.1 μg/mL of TM, cytotoxicity was relatively decreased and GRP78 and GRP94 expression levels were significantly increased (Figure 2). Moreover, there was no obvious change in intracellular ROS accumulation and apoptosis at the same concentrations (data not shown). Similar ranges of TG or TM concentration and a specific timepoint were used to induce ER stress proteins without serious toxicity in various cell lines [13,14,31]. However, prolonged exposure resulted in decreased GRP78 and GRP94 levels, leading to a consequent loss of cell viability. Taken together, these findings suggest that GRP78 and GRP94 may be induced in cells prior to more severe cytotoxic development.

ER stress exposure resulted in either activation of protective ER stress responses or ER-associated apoptotic pathways, which occur due to ER stress beyond the capacity of the UPR system. The protective response restored cellular homeostasis and adaptive reactions that potentiate protective abilities against a later and more injurious stress. The beneficial effect of mild ER stress, assimilated as ER stress preconditioning, has been reported in the liver or brain of TM-injected rats, which were protected from later hepatic ischemia/reperfusion injury or lipopolysaccharide-induced neuroinflammation and memory impairment, respectively [32,33]. Additionally, ER stress preconditioning in cultured cells pretreated with TG alleviated toxicant-mediated cell damage through the upregulation of ER stress-related proteins, including GRP78 and GRP94 [14,34]. In the present study, pre-incubation of HEI-OC1 cells with TG or TM prior to adding cisplatin induced GRP78 and GRP94 expression, attenuated intracellular ROS accumulation, and inhibited the caspase-dependent apoptotic pathway, resulting in increased cell viability (Figure 3), which were correlated with an amelioration of TUNEL-positive cells (Figure 4). These results define a novel mechanism wherein mild ER stress may be beneficial for auditory cells in defending against the ototoxic side effect of cisplatin, as it can alleviate cell injury, including excessive ROS accumulation and apoptosis.

GRP78 and GRP94 induction in ER stress-preconditioned cells plays cytoprotective roles in various cytotoxic conditions. For example, increased GRP78 expression during ER stress response attenuated H_2_O_2_-induced renal epithelial cell injury by inhibiting the increase of intracellular Ca^2+^ concentration and activation of the ERK1/2 signaling pathway [35]. Tolerance to various cytotoxins was provided by GRP78 and GRP94 overexpression in several cell lines [31]. Furthermore, the co-downregulation of GRP78 and GRP94 expressions in prostate cancer cells by their specific siRNAs suppressed cell migration and promoted caspase-9-dependent apoptosis [36]. In the present study, transient transfection of HEI-OC1 cells with siRNA targeted against GRP78 or GRP94 abolished both inducers’ expression levels during ER stress preconditioning and failed to reduce cisplatin-mediated ROS accumulation, thus sensitizing cells to cisplatin-induced cytotoxicity (Figure 5). This finding indicated that GRP78 and GRP94 induction is integral for ER function to promote a protective mechanism against cisplatin-triggered ototoxicity. This is supported by recent findings that the decreased expression of ER stress-related proteins, including GRP78, in the cochleae of aged mice was associated with age-related hearing loss [16], and that intense noise exposure upregulated GRP78 expression level in hair, lateral wall, and spiral ganglion cells of guinea pigs, thereby protecting cochlear cells from noise-induced injury [17]. It has been also reported that cisplatin binds to GRP78 and GRP94 from cochlear and kidney cell lysates, suggesting that this interaction may attenuate cisplatin ototoxicity [37]. It will be of interest to examine the signaling pathways of ER stress sensors, including inositol-requiring enzyme 1 (IRE1), PKR-like endoplasmic reticulum kinase (PERK), and activating transcription factor 6 (ATF6). This may help to understand the protective mechanism of ER stress preconditioning against ototoxicity of cisplatin.

## 5. Conclusions

In summary, we showed that TG or TM pretreatment before cisplatin exposure attenuated cisplatin ototoxicity in auditory hair cells. The protective effects of these ER stress inducers were achieved through increasing GRP78 and GRP94 expressions, leading to the inhibition of intracellular ROS accumulation and activation of intrinsic apoptotic signaling pathways induced by cisplatin. Our findings add to the knowledge of the beneficial effects of ER stress preconditioning on cisplatin-induced ototoxicity and also provide new insight in designing approaches to prevent or treat environment-related hearing loss.

## Figures and Tables

**Figure 1 antioxidants-09-00686-f001:**
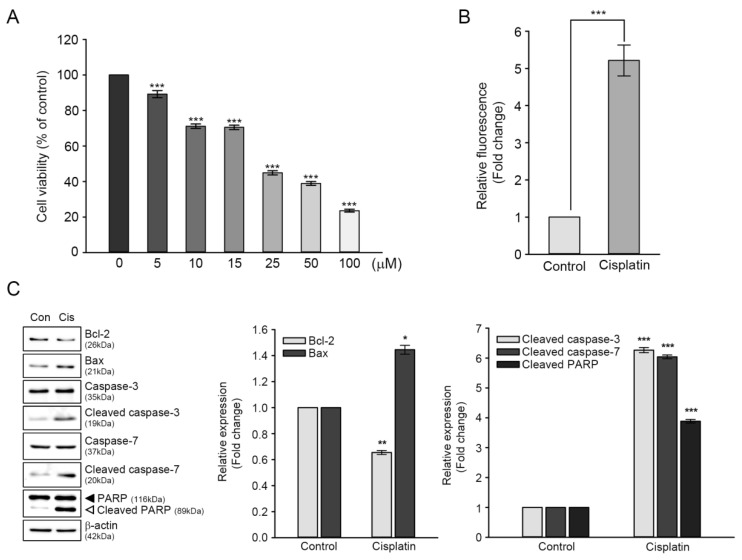
Effects of cisplatin on viability, reactive oxygen species (ROS) generation, and expression of apoptosis-related proteins of HEI-OC1 cells. Cells were treated with various concentrations of cisplatin (5–100 μM) for 24 h. (**A**) Cell viability was determined through CCK assay. Values are expressed as means ± SE of four independent experiments, expressed as a percentage of untreated control values. *** *p* < 0.001, compared with untreated control. (**B**) Intracellular ROS accumulation was measured through DCF fluorescence intensity spectrofluorometry. Values are expressed as means ± SE of four independent experiments, expressed as a percentage of untreated control values. *** *p* < 0.001, compared with untreated control. (**C**) Representative immunoblotting result showing the expression of apoptosis-related proteins. After treatment with 25 μM of cisplatin for 24 h, the cells were harvested for immunoblotting of each protein. Protein bands were quantified using densitometry, and their abundances were expressed relative to β-actin band density. The ratio of each protein to β-actin is presented as a fold change of that of the untreated control. Values are expressed as means ± SE of three independent experiments. * *p* < 0.05, ** *p* < 0.01, *** *p* < 0.001; compared with the untreated control.

**Figure 2 antioxidants-09-00686-f002:**
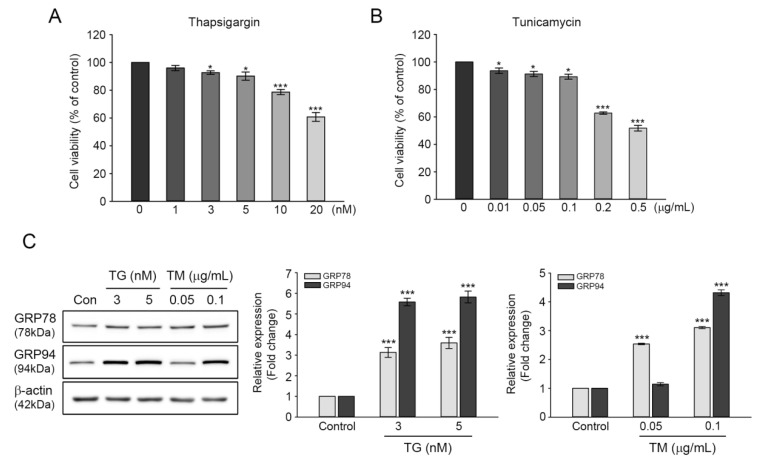
Induction of GRP78 and GRP94 expression in thapsigargin (TG)- or tunicamycin (TM)-treated HEI-OC1 cells. (**A**) Cytotoxicity was mediated by 1–20 nM TG (**A**) or 0.01–0.5 μg/mL TM (**B**) treatment for 24 h. Values are expressed as means ± SE of four independent experiments, expressed as a percentage of untreated control values. * *p* < 0.05, *** *p* < 0.001, compared with the untreated control. (**C**) Cells were treated with 3 and 5 nM TG, or 0.05 and 0.1 μg/mL TM for 24 h, followed by immunoblot of GRP78 and GRP94 expressions. Individual bands were quantified densitometrically and normalized to β-actin. The values in a graph are represented as fold changes relative to the untreated control, expressed as means ± SE of three independent experiments (*** *p* < 0.001).

**Figure 3 antioxidants-09-00686-f003:**
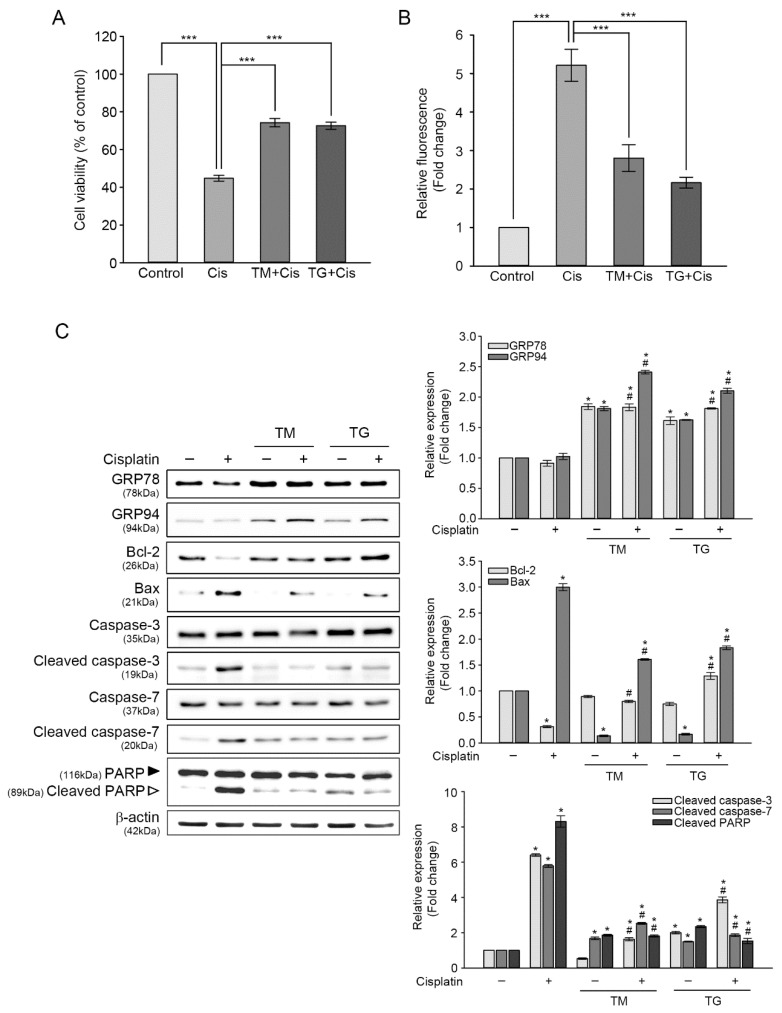
Protective effect of GRP78 and GRP94 induction on cisplatin-mediated cytotoxicity. (**A**) Cells were pretreated with 0.1 μg/mL of TM or 3 nM of TG or for 24 h and then treated with 25 μM of cisplatin for another 24 h. (**A**) Cell viability was measured through CCK assay. Values are expressed as means ± SE of four independent experiments, expressed as a percentage of untreated control values. *** *p* < 0.001, compared with untreated control. (**B**) ROS levels were determined through DCF fluorescence intensity spectrofluorometry. Values are expressed as means ± SE of three independent experiments. *** *p* < 0.001, compared with the untreated control. (**C**) Expressions of apoptosis-related proteins were analyzed through immunoblotting. β-actin was used as a control for protein loading. The graph represents relative protein levels compared with the untreated control after normalization to β-actin levels. Values are expressed as means ± SE of three independent experiments. *^,#^
*p* < 0.05, *; compared with the untreated control, ^#^; cisplatin-only versus TM plus cisplatin or TG plus cisplatin.

**Figure 4 antioxidants-09-00686-f004:**
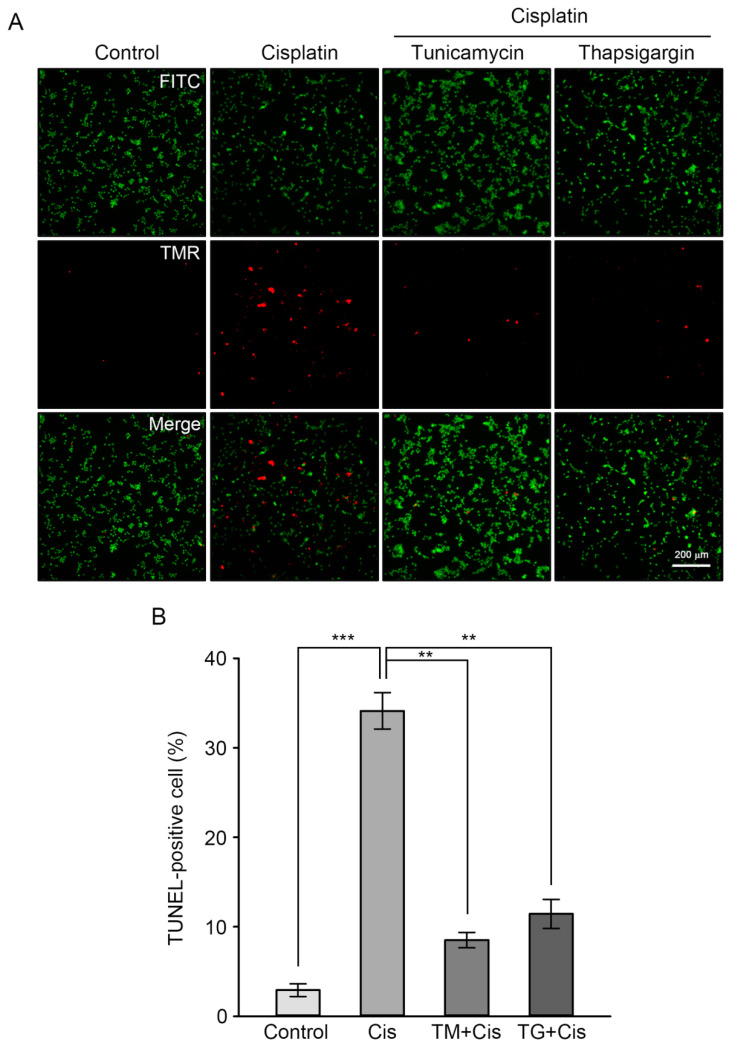
Attenuation of cisplatin-mediated apoptosis through GRP78 and GRP94 overexpression. Cells were treated with TM or TG and then cisplatin, as previously described. (**A**) Fluorescence images with calcein AM were examined to detect “live” cells (FITC; green) and “apoptotic” cells (TMR; red). Scale bars = 200 μm. Original magnification, 40×. (**B**) The percentage of TUNEL-positive cells as shown in (**A**) was determined from ≈1000 cells that spanned 3–4 microscopic fields. Values are expressed as means ± SE. ** *p* < 0.01, compared with cisplatin-treated cells; *** *p* < 0.001, compared with the untreated control.

**Figure 5 antioxidants-09-00686-f005:**
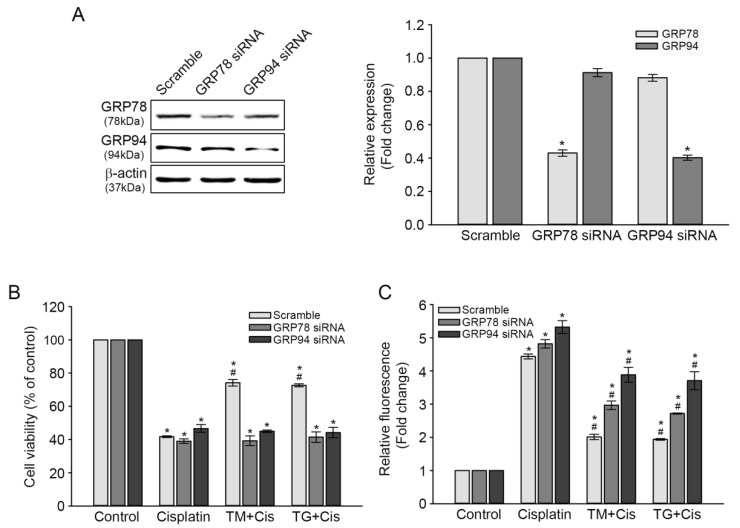
Cisplatin-mediated cytotoxicity and intracellular ROS accumulation in GRP78 or GRP94 KD cells. Cells were transfected with GRP78, GRP94, or scrambled siRNAs and then treated with cisplatin as described in Section 2. (**A**) GRP78 and GRP94 expression levels after the 72-h transfection. Protein bands were quantified densitometrically and normalized to the density of the β-actin band. The ratio of GRP78 or GRP94 to β-actin in each group was presented as its fold-change relative to the scrambled siRNA transfectant. * *p* < 0.05, compared with the scrambled siRNA transfectant. After the 48-h incubation, cells were treated with TM or TG, and then cisplatin, as previously described. (**B**) Cell viability was determined through CCK assay. The graph represents the relative viability percentage, compared with the untreated control. Values are expressed as means ± SE of three independent experiments. *p* < 0.05, *; compared with the untreated control, ^#^; cisplatin-only versus TM plus cisplatin or TG plus cisplatin. (**C**) The levels of ROS accumulation were determined through DCF fluorescence intensity spectrofluorometry. The graph represents the relative ROS accumulation fold, compared with untreated controls. Values are expressed as means ± SE of three independent experiments. *p* < 0.05, *; compared with the untreated control, ^#^; cisplatin-only versus TM plus cisplatin or TG plus cisplatin.

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
