# Peer review of "Protective Effects of Glucose-Related Protein 78 and 94 on Cisplatin-Mediated Ototoxicity"

_antioxidants, 2020, doi:10.3390/antiox9080686_

Round 1

Reviewer 1 Report

The authors analyzed the role of ER chaperone proteins Grp78 and Grp94 expression on cisplatin-induced cytotoxicity/ ototoxicity in vitro. In addition, they have shown that thapsigargin and tunicamycin reduce cisplatin-mediated cytotoxicity of auditory cells, which was associated with an upregulation of Grp78 and Grp94. The study is well designed and structured. The reader can easily follow the authors' explanations. The results and figures are clearly described. I recommend the manuscript for acceptance.

Author Response

We highly appreciate your comments with honor.

Reviewer 2 Report

The manuscript produced is very interesting and well written. I believe it can be published

Author Response

(The authors gave the same response as above.)

Reviewer 3 Report

Cisplatin is a chemotherapeutic drug which ototoxic properties that can damage the cochlea and lead to hearing loss. Unfortunately, these adverse events cannot be cured or prevented. In this manuscript, Yi et al. used murine auditory cells (HEI-OC1) and cisplatin to mirror ototoxicity. Application of cisplatin to HEI-OC1 cells decreased cell viability, increased reactive oxygen species (ROS) and increased proteins involved in apoptosis. However, pre-incubation of HEI-OC1 cells with ER stressors thapsigargin or tunicamycin upregulated GRP78 and GRP94, regulators of the Unfolded Protein Response, which minimized Cisplatin-driven cell death, ROS and apoptosis pathway activation. The authors confirmed the protective effects of GRP78 and GRP94 using a knockdown approach. This work indicates that GRP activation can attenuate cisplatin-mediated ototoxicity. This study is very well done and the experiments are both comprehensive as well as logical. The authors have used multiple innovative techniques to investigate the interaction of GRP and cisplatin and the conclusions are well supported by the findings. As a result, I have only minor comments. 

Minor Comments:

  1. Please include the cat #s for the antibodies used in the methods.
  2. In the methods section, Line 143, please indicate the amount of time the cells were permeabilized.
  3. In line 170, please add the ± stdev in the text when citing the change in viability (%).
  4. In line 1717, I recommend replacing the word “furthermore” with “as a result”.
  5. In Figure 1C, the levels of GRP78 do not appear to be different between the groups, but the quantification indicates statistically significant elevation in response to tunicamycin and thapsigargin. Please elucidate.
  6. GPR78 is upstream of IRE1, ATF6 and elF2. The discussion would benefit from speculation on how pre-treatment of cells with ER stressor likely activate these systems and how they could potentially contribute to attenuation of ototoxicity.

Author Response

Minor Comments:

  1. Please include the cat #s for the antibodies used in the methods.

: As suggested, cat #s of all primary antibodies used in the experiments have been inserted next to the name of each primary antibody mentioned in ‘Materials’ subsection.

  1. In the methods section, Line 143, please indicate the amount of time the cells were permeabilized.

: As suggested, the incubation time for permeabilization has been inserted in methods section (p4, ln145).

  1. In line 170, please add the ± stdev in the text when citing the change in viability (%).

: As suggested, the ±SE has been added to ‘Figure 1 legend’ (p5, ln180-182) with four repetitive experiments.

  1. In line 1717, I recommend replacing the word “furthermore” with “as a result”.

: As suggested, ‘furthermore’ has been replaced with ‘as a result’ in the results 3.1 subsection (p4, ln173).

  1. In Figure 1C, the levels of GRP78 do not appear to be different between the groups, but the quantification indicates statistically significant elevation in response to tunicamycin and thapsigargin. Please elucidate.

: Fig. 1C represents the immunoblot for the changes in expressions of apoptosis-related proteins mediated by cisplatin. We guess the figure you mentioned is Fig. 2C. The level of GRP78 expression was quantified using a chemidoc from GE Health Care (LAS 500 Biomolecular Imager and its quantification software) and normalized with that of β-actin as a loading control. Immunoblotting was independently repeated three times with similar values. This result is shown in Fig. 2C.

  1. GPR78 is upstream of IRE1, ATF6 and elF2. The discussion would benefit from speculation on how pre-treatment of cells with ER stressor likely activate these systems and how they could potentially contribute to attenuation of ototoxicity.

: This will be our further study. Preconditioning effect of ER stress sensors on cisplatin ototoxicity has been briefly discussed in Discussion section (p12, ln382-385).